# Monitoring Carbapenem-Resistant Enterobacterales in the Environment to Assess the Spread in the Community

**DOI:** 10.3390/antibiotics11070917

**Published:** 2022-07-08

**Authors:** Taro Urase, Saki Goto, Mio Sato

**Affiliations:** School of Bioscience and Biotechnology, Tokyo University of Technology, Tokyo 192-0982, Japan; gotohsk@stf.teu.ac.jp (S.G.); sugar122527@yahoo.co.jp (M.S.)

**Keywords:** Enterobacterales, carbapenem-resistant, carbapenemase, treated wastewater, one health, wastewater-based epidemiology

## Abstract

The usefulness of wastewater-based epidemiology (WBE) was proven during the COVID-19 pandemic, and the role of environmental monitoring of emerging infectious diseases has been recognized. In this study, the prevalence of carbapenem-resistant Enterobacterales (CRE) in Japanese environmental samples was measured in the context of applying WBE to CRE. A total of 247 carbapenem-resistant isolates were obtained from wastewater, treated wastewater, and river water. Treated wastewater was shown to be an efficient target for monitoring CRE. The results of the isolate analysis showed that WBE may be applicable to *Escherichia coli*-carrying New Delhi metallo-β-lactamase (NDM)-type carbapenemase, the *Enterobacter cloacae* complex and *Klebsiella pneumoniae* complex-carrying IMP-type carbapenemase. In addition, a certain number of CRE isolated in this study carried Guiana extended spectrum (GES)-type carbapenemase although their clinical importance was unclear. Only a few isolates of *Klebsiella aerogenes* were obtained from environmental samples in spite of their frequent detection in clinical isolates. Neither the KPC-type, the oxacillinase (OXA)-type nor the VIM-type of carbapenemase was detected in the CRE, which reflected a low regional prevalence. These results indicated the expectation and the limitation of applying WBE to CRE.

## 1. Introduction

Wastewater-based epidemiology (WBE) served as an important tool during the COVID-19 pandemic to track the circulation of the pathogen in a community [1]. Monitoring SARS-CoV-2 RNA in wastewater provided an early warning for reemergence independent of patient testing or hospital reporting and includes data on asymptomatic individuals [2]. Successful application of WBE in tracking the pandemic viral outbreak suggests the possibility of its application to emerging pathogens that cause not only gastrointestinal but also respiratory infections.

Carbapenems are antibiotics that work for people affected by severe infectious diseases caused by multidrug-resistant Gram-negative bacteria, such as Enterobacterales, *Acinetobacter baumannii* and *Pseudomonas aeruginosa* [3]. Because of the importance of carbapenems in chemotherapy, carbapenem-resistant Enterobacteriaceae is listed by the WHO as a critically important bacterial group for which new antibiotics are urgently needed [4]. The importance of the one-health approach has been recognized in the management of carbapenem-resistant Enterobacterales (CRE) [5,6,7], because not only hospital wastewater but municipal and even treated wastewater have been reported to contain CRE. A study of U.S. wastewater showed that 65 out of 322 *Escherichia coli* isolates were imipenem-resistant [8], but the reported incidence of CRE was kept in a low range of 0.3–2.93 infections per 100,000 person-years across the entire U.S. population [9]. Although a high removal (>99.9%) of CRE from wastewater was reported in the treatment processes [10], the treatment plants are considered to be hotspots for resistant bacteria in the environment [11].

During quantitative monitoring, which is essential for WBE, the genes that encode carbapenemases are usually quantification targets because of their ubiquitous presence [12,13]. They are carried by various species under Enterobacterales such as *Klebsiella quasipneumoniae*, *Escherichia coli*, and *Enterobacter cloacae,* which were taken from municipal wastewater samples [14]. Molecular-based techniques such as new generation sequencing and real-time polymerase chain reaction (PCR) tests have been used for the quantitative monitoring of carbapenemase genes [12,15]. Carbapenemases constitute a large variety of enzymes that are categorized into three of the four Ambler classes (A, B and D) [13]. The major members of class A are *Klebsiella pneumoniae* carbapenemase (KPC) and Guiana extended spectrum (GES). Members of class B are metallo-β-lactamases (MBLs), consisting mainly of imipenem-resistant *Pseudomonas* (IMP); Verona integron-encoded metallo-β-lactamase (VIM); and New Delhi metallo-β-lactamase (NDM). The most prevalent member of class D is oxacillinase (OXA). These are spreading over the world at alarming rates [16].

In addition to molecular-based approaches, culture-based methods are still useful for analyzing isolates to obtain information on the linkage with pathogens. However, the number of reports on the quantitative detection of CRE in environmental samples is limited due to the presence of carbapenem-resistant bacteria other than Enterobacterales. Incubation at a higher temperature (42 °C) was effective for the selective growth of CRE in environmental samples which generally contain autochthonous species that have an intrinsic resistance to carbapenems, such as *Stenotrophomonas maltophilia* [17,18]. A high dominance of *Aeromonas* spp. in a carbapenem-resistant bacterial population [19] is also a problem for the selective isolation of CRE.

This study aimed to clarify the CRE species detectable by environmental monitoring with a careful comparison in the prevalence of carbapenemase genes in environmental isolates with those in clinical isolates. The potential and limitations of applying WBE to CRE are discussed.

## 2. Results

### 2.1. Prevalence of CRE

Figure 1 shows the results of colony counts at an incubation temperature of 42 °C. High plate-count numbers were observed for B1 raw wastewater samples. Except for B1 samples, treated wastewater (A1–A4) and river water (TM, TR, K3, K7, K11) contained Enterobacterales in a range of 4–1750 CFU/mL. The number of colonies on the meropenem-containing plates were in the range of ND (no colony in 100 mL) and 56 CFU/mL. The ratios of presumptive CRE to total Enterobacterales were between 0.01 and 1% for all samples, excluding the cases of ND. Although the bacterial numbers were high in the raw wastewater samples, the ratios of presumptive CRE to total Enterobacterales were below 0.02%. The ratio of CRE numbers to the heterotrophic plate count numbers was reported to be approximately 10^−4^ (0.01%) to 10^−5^ (0.001%) for wastewater samples in previous studies [10,18]. In our previous study, the ratios of presumptive CRE were observed between 0.014 and 0.31% using slightly different isolation methods from those in this study [20]. Focusing the target species on *Escherichia coli*, resistance ratios of 0.8% for hospital effluent and 0.3% for wastewater treatment plant influent were reported in Japan [21]. The variations in the reported presumptive CRE ratios based on colony counts were relatively small in spite of regional and methodological differences.

Figure 2 shows the effect of the addition of zinc sulfate (ZnSO_4_) on the number of colonies under the selection pressure of meropenem. The addition of ZnSO_4_ at a concentration of 7 mg/L increased the number of colonies, while 70 mg/L decreased the number. To obtain more CRE isolates, including producers of metallo-β-lactamase (MBL) from environmental samples, 7 mg/L was considered to be an appropriate dose.

### 2.2. Confirmation of Meropenem Resistance

A total of 243 isolates (66%) out of 367 examined colonies obtained at 37 °C incubation were resistant to meropenem by the disk diffusion test (Table 1). A total of 206 isolates (84%) out of 243 meropenem-resistant colonies showed MBL production by the double-disk synergy test. As will be mentioned in a later section, these isolates, originating from 37 °C incubation, were often identified as *Stenotrophomonas maltophilia*, an autochthonous species with intrinsic resistance to carbapenems. In the case of isolates obtained at 42 °C incubation, 128 of 238 isolates (54%) were resistant to meropenem. MBL production was seen in 73 of 128 meropenem-resistant isolates (57%).

### 2.3. Identification of Species of the Meropenem-Resistant Isolates

Among 371 meropenem-resistant isolates, the species of randomly selected 247 isolates were identified by 16S rRNA gene sequencing (Accession numbers are shown in Appendix A). The results were summarized into the genus level in Table 2 totaled by each sampling location. One isolate from river water was identified as a member of Enterobacterales. The raw wastewater contained CRE at a low ratio due to high dominance of *Stenotrophomonas* spp. and *Psedomonas* spp., and a majority of CRE isolates were obtained from treated wastewater samples. Table 3 shows the results of species identification by each incubation method. More than half (68%) of the isolates obtained at 37 °C without zinc addition were identified as bacteria other than Enterobacterales, such as *Stenotrophomonas maltophilia*, although the zinc addition improved the CRE ratio. For isolates obtained at 42 °C without zinc addition, the CRE ratio increased up to 70%. An increase was also found for the isolates obtained at 42 °C with zinc addition. The better CRE selection under incubation at the higher temperature was consistent with the literature [18]. This study confirmed a better CRE selection with zinc from environmental samples, while it had been shown for clinical isolates with a different addition dose of zinc [22].

In total, 51% of the examined meropenem-resistant isolates belonged to Enterobacterales, while the residual 49% belonged to other orders, such as *Stenotrophomonas* spp., *Pseudomonas* spp., and *Aeromonas* spp. Among Enterobacterales, high dominance was observed for *Enterobacter* spp., followed by *Klebsiella* spp.

The result of species-level identification is shown in Figure 3. The isolates under the *Enterobacter* genus consisted of *E. quasiroggenkampii*, *E. ludwigii*, *E. kobei*, *E. asburiae*, *E. cloacae*, *E. sichuanensis*, and *E. huaxiensis*. Among these species, *E. ludwigii*, *E. Kobei*, *E. asburiae*, *E. cloacae* were categorized into *E. cloacae* complex in clinical examination [23]. *E. quasiroggenkampii*, as a high-population species of in this study, was also close to other *E. cloacae* complex members [24]. The five isolates identified as *E. sichuanensis* (the closest accession POVL01000141) had only 6–9 variations in more than 1460 nucleotide sequences compared with those of *E. ludwigii* (accession JTLO01000001) or *E. Kobei* (accession CP017181). Only one isolate identified as *E. huaxiensis* (the closest accession MK049964) had a similarity less than 98.65% to any accessions of known *E. cloacae* complex isolates. These results indicated that most of the isolates identified as *Enterobacter* spp. belonged to the *E. cloacae* complex and related species.

Among the identified isolates in *Klebsiella* spp., *K. pneumoniae*, *K. quasipneumoniae*, *K. variicola,* and *K. quasivariicola* were categorized into a *K. pneumoniae* complex [25]. Only three isolates were identified as *K. aerogenes*, an important species in CRE. Seven isolates were identified as *Raoultella ornithinolytica* (the closest accession was AJ251467). These seven isolates had a relatively small number (13–17) of variations in more than 1460 nucleotide sequences (more than 98.9–99.2% similarity) compared with those of *K. aerogenes* (Accession CP002824). The dominance of *Escherichia/Shigella* spp. and *Citrobacter freundii* was relatively small.

These populations of CRE were consistent with a result from Japanese and Taiwanese wastewater monitoring in which *Klebsiella quasipneumoniae* were isolated as carbapenemase-producing Enterobacterales (CPE) followed by *Escherichia coli*, *Enterobacter cloacae* complex, *Klebsiella pneumoniae*, *Klebsiella variicola*, *Raoultella ornithinolytica*, *Citrobacter freundii*, and *Citrobacter amalonaticus* [14].

Although *Stenotrophomonas*, *Pseudomonas* and *Aeromonas* are not members of Enterobacterales, they sometimes form red-colored colonies on ECC agar plates, which interferes with detection of CRE. A high dominance of *Stenotrophomonas maltophilia*, which shows intrinsic resistance to carbapenems [18], was observed, followed by *Pseudomonas* spp. and *Aeromonas* spp. among members outside Enterobacterales. A high dominance of *Aeromonas* spp. in the population of Gram-negative carbapenem-resistant bacteria was consistent with the literature on wastewater samples [19].

### 2.4. Carbapenemase-Producing Genes Carried by the Isolates

Figure 4 shows the carbapenemase-producing genes carried by the isolates. A high prevalence of GES carried by *Enterobacter*, *Klebsiella* and *Raoultella* was detected. Detection of GES was highly reliable because in each case it was confirmed by PCR with different primers to distinguish GES-type carbapenemase from GES with ESBL-like activity as shown in the materials and method section. The isolates of *Enterobacter* spp. carried the GES-type carbapenemase with the highest dominance (27 isolates out of 58), followed by GES with ESBL-like activity, IMP-1 and IMP-6, while no target genes were detected for 13 isolates. GES genes were detected only from isolates of *E*. *quasiroggenkampii* although the clinical importance of *Enterobacter* spp. carrying GES is unclear. The isolates of *Klebsiella* spp. carried the GES-type carbapenemase with the highest dominance (24 isolates out of 47), followed by IMP-6 and IMP-1. A total of 6 out of 9 *Escherichia*/*Shigella* isolates carried NDM. All isolates of *Raoultella* carried the GES-type. Two isolates of *Citrobacter* carried IMP (IMP-1 or IMP-6). Neither KPC, OXA nor VIM was detected.

Table 4 shows the results of carbapenemase genes carried by Enterobacterales with different isolation conditions. Although the number of isolates with zinc addition was limited, high ratios (57–63%) of isolates with metallo-β-lactamases (IMP-1, IMP-6 and NDM) were obtained by the addition of zinc to the incubation medium.

## 3. Discussion

### 3.1. Species Monitorable from Environmental Samples

A 2019 surveillance program in Japan indicated that the CRE isolated in Japan consisted of *Klebsiella aerogenes* (40.7%), *Enterobacter cloacae* complex (29.6%), *Klebsiella pneumoniae* (10.0%), *Escherichia coli* (6.5%), *Serratia marcescens* (3.4%), and *Citrobacter freundi* (2.1%) [26]. Figure 5 was drawn based on another Japanese surveillance (JANIS), which reported the number of meropenem-resistant Enterobacterales together with the resistance ratios to meropenem [27,28]. The numbers in this figure are the totals of the examinations both for inpatients [27] and outpatients [28]. The figure also indicates the ratios of meropenem-resistance for all examined isolates. The resistant ratios to meropenem were less than 1% regardless of species, which indicated that CRE prevalence had been kept low. A comparison of Figure 5 with Figure 3 shows that *Enterobacter* and *Klebsiella* were both representative genera of carbapenem resistance in the case both of clinical and environmental isolates. However, a species-level difference was found: the number of *Klebsiella aerogenes* from the environmental samples was clearly smaller than expected from the surveillance on nosocomial infections [27,28]. It must be noted that the isolates identified as *Raoultella ornithinolytica* might be identified as *K. aerogenes* in a routine clinical examination because of the strong similarity (98.9–99.2%) in 16S rRNA sequencing results. Another reason for the lower detection of *Klebsiella aerogenes* might be that they rarely carried carbapenemase genes [29]. On the other hand, considering the use of WBE to CRE, the prevalence of carbapenem resistance in the *Enterobacter cloacae* and *Klebsiella pneumoniae* complexes could be monitored by environmental samples. 

### 3.2. Carbapenemase Carried by Environmental Isolates

A high prevalence of GES-type carbapenemase was observed among isolates of *Enterobacter*, *Klebsiella* and *Raoultella* although detection of GES is relatively rare in clinical isolates [26,30]. Although the reason is not clear, one probable reason for the apparent disparity between the resistome of clinical and wastewater environments might be the inefficient detection of GES-5 in clinical routine analysis due to the low carbapenem resistance [31]. Both IMP-type and GES-type carbapenemases were simultaneously found from an isolate of *E. quasiroggenkampii* and *Klebsiella* spp. which might harbor these genes by the same mechanism as recently reported in Japan [32].

This study indicated that CRE-carrying IMP-type carbapenemase, which is considered to be the dominant carbapenemase in Japan [26,29], could be monitored by environmental monitoring, judging from the high detection frequency of IMP-type carbapenemase carried by the *Enterobacter cloacae* and *Klebsiella pneumoniae* complexes. On the other hand, no isolates carried KPC, although its high dominance in clinical isolates of *Klebsiella pneumoniae* had been reported in China [33,34,35]. The fewer incidences of KPC detection could be the result of the characteristics of environmental samples [31], the expected high removal of KPC in wastewater treatment [12], and the small population of KPC in clinical isolates [26,29,36] although the reason may be complex.

The detection of NDM as a dominant carbapenemase in *Escherichia*/*Shigella* isolates is consistent with its prevalence in Japan [37] and other east Asian countries [35,38]. Although in European countries NDM is spreading to *Citrobacter freundii* and the *Enterobacter cloacae* complex [30], only one isolate carried NDM by *K. quasipneumoniae* in this study, apart from the *Escherichia*/*Shigella* isolates. Although OXA is spreading among Enterobacterales in Europe and northern Africa [39,40,41,42], no isolates carrying OXA were detected, reflecting the regional prevalence in clinical isolates [26]. The regional difference in the prevalent types of carbapenemases was clearly seen from environmental monitoring.

### 3.3. Effect of Isolation on the Detected Type of Carbapenemase

The ratio of Enterobacterales among the isolates increased with the addition of zinc at the higher incubation temperature (42 °C). In addition, high ratios (57–63%) of isolates with metallo-β-lactamases (IMP-1, IMP-6 and NDM) were obtained by adding zinc to the incubation medium, as described in Section 2. This result was reasonable because the objective of adding zinc was to select more isolates with metallo-β-lactamases [22]. The incubation conditions (temperature and composition of the medium) are essential in wastewater-based epidemiology. Careful optimization of these parameters may be needed depending on the type of carbapenemase genes in the target isolates and the presence of autochthonous species with intrinsic resistance to carbapenems.

## 4. Materials and Methods

### 4.1. Sample Origins

Treated wastewater samples after chlorination at four wastewater treatment plants (WTPs) and raw wastewater samples at one WTP were collected in 2019–2021 on days with limited rain events (Table 5), except for samples of two isolates taken in 2018 (detailed information is given in Appendix A). All of the treatment plants, in Tokyo and its suburbs (Figure 6), are operated by municipalities. They treat domestic and some industrial and hospital wastewater by the activated sludge process with a modified anaerobic-anoxic-aerobic (A_2_O) operation. In addition, river water with the characteristics shown in the same table were also collected in the Tokyo metropolitan area and in Kanagawa prefecture. The water samples were collected in sterilized glass bottles in the morning, and immediately brought back to the laboratory for analysis in the afternoon the same day.

### 4.2. Isolation of CRE

An appropriate volume (0.1 to 100 mL) of a water sample was filtered through a nitrocellulose membrane of 47 mm diameter with 0.45 µm pore size (Millipore, Billerica, MA, USA) to collect bacteria contained in the sample. The membranes with several steps of sample volume were prepared to obtain countable numbers of colonies on agar plates. The membrane filter was then placed on the agar plate made from a selective chromogenic medium CHROMagar ECC (CHROMagar, Paris, France) that allowed simultaneous detection and differentiation between *Escherichia coli* and other Enterobacterales by colony coloration [43] with and without meropenem (FUJIFILM Wako Pure Chemical Corporation, Osaka, Japan).

In 2010, the Clinical and Laboratory Standards Institute (CLSI) lowered the minimum inhibitory concentration (MIC) breakpoints for resistant Enterobacteriaceae to carbapenems (e.g., from 16 to 4 μg/mL for imipenem and meropenem) [44]. The European Committee on Antimicrobial Susceptibility Testing (EUCAST) distributes MIC breakpoint tables for resistant Enterobacteriaceae that are slightly different from those by CLSI (e.g.: 4 μg/mL for imipenem and 8 μg/mL for meropenem) [45]. The concentration of meropenem added to the medium was set at 0.5 μg/mL in this study, which was lower than the breakpoints provided by CLSI and EUCAST. The reason for the lower concentration was to select more presumptive colonies based on our previous results [20] so that CRE would be selected by the confirmation test from the presumptive colonies [10].

Regarding the analysis of a part of the samples (A1-November-2021 and A3-December-2021), ZnSO_4_ was added to the culture medium at concentrations of 7 and 70 mg/L on the isolation of CRE. The objective was to improve the isolation by promoting the growth of CRE-harboring MBL, as shown in the literature [22].

The plates with membranes were incubated at 37 and 42 °C for 24 h. The objective of the 42 °C condition was to obtain CRE efficiently because autochthonous species with intrinsic resistance to carbapenems, such as *Stenotrophomonas maltophilia*, often form more colonies than those of the target carbapenem-resistant bacteria on the carbapenem-containing agar plates [18,46]. The number of blue and red colonies on the agar plates with and without meropenem were counted as the numbers of Enterobacterales and presumptive CRE, respectively. Several colonies were randomly picked up and streaked onto other plates under the same selective pressure again to obtain isolates.

### 4.3. Confirmation of Resistance for Presumptive CRE

The disk diffusion test based on the Kirby–Bauer method was conducted to identify the susceptibility of isolates to meropenem by using agar plates of Mueller–Hinton medium (Eiken Chemical Co., Ltd., Tokyo, Japan). The antimicrobial disks containing meropenem (meropenem; 10 μg) (KB disk, Eiken Chemical Co., Ltd., Tokyo, Japan) were used according to the manufacturer’s instruction. Susceptibility was interpreted as susceptible (S), intermediate (I), or resistant (R) based on the comparisons of the inhibition zone diameters (S: 23 mm or more, I: 20–22 mm, R: 19 mm or less) around the diffusion disks with the criteria [44]. A disk containing sodium mercaptoacetic acid (SMA, 3 mg, Eiken Chemical Co., Ltd., Tokyo, Japan) was also used in the double-disk synergy test to determine MBL production by the presumptive CRE [47].

### 4.4. Identification of Species

The phylogenetic characteristics of the isolates were examined by sequencing full-length 16S rRNA gene after confirmation of their resistance to meropenem. Before sequencing, DNA was extracted by using a DNA extraction kit (Kanto Chemical Co., Ltd., Tokyo, Japan). The extracted 16S rRNA gene was amplified by primers 27F (5′-AGAGTTTGATCMTGGCTCAG-3′) and 1492R (5′-GGYTACCTTGTTACGACTT-3′). The amplified DNA was purified by a MonoFas DNA clean-up kit (GL Sciences Inc., Tokyo, Japan) and sent to Macrogen Japan Corp. (Tokyo, Japan) for sanger sequencing with primers of 518F (5′-CCAGCAGCCGCGGTAATACG-3′) and 800R (5′-TACCAGGGTATCTAATCC-3′). The two obtained sequences were assembled to give a full-length 16S rRNA sequence. The closest species or genus above 98.65% of the identity [48] was determined by a comparison with type strains from the BLASTn program (https://blast.ncbi.nlm.nih.gov/ (accessed on 31 January 2022)) provided by National Center for Biotechnology Information (NCBI), National Institute of Health, U.S.

### 4.5. Genotyping of Carbapenemase Genes

A PCR screening test was performed to identify the genes (IMP-type, NDM-type, VIM-type, KPC-type, OXA-48-like-type, and GES-β-lactamase (GES)-type) encoding carbapenemase by using a kit (Kanto Chemical Co., Ltd., Tokyo, Japan) based on the literature [49]. To distinguish GES with carbapenemase-like activity from GES with ESBL-like activity, an additional PCR test was conducted using primers GES-f-v1 (5’-TCCCCAAGGAGAGATCGTCG-3’), GES-r-v1 (5’-TCGCCAGGTGTGTTGTCGTT-3’), GES-R (5’-CCTCTCAATGGTGTGGGT-3’) based on a previous study [50].

## 5. Conclusions

The results of the analysis on 247 carbapenem-resistant isolates indicated that *Escherichia coli*-carrying NDM-type carbapenemase, and the *Enterobacter cloacae* and *Klebsiella pneumoniae* complex-carrying IMP- and GES-types could be monitored from environmental samples even including treated wastewater. The concept of wastewater-based epidemiology can be applied to the monitoring of carbapenem-resistant Enterobacterales of these species although the clinical significance of a large number of isolates carrying the GES-type was unclear. A lower number of *K. aerogenes* isolates than expected from surveillance programs implied that it may be a difficult species to monitor from wastewater samples although it was suggested that *Raoultella ornithinolytica* in this study might be identified as *K. aerogenes* in a routine clinical examination. No isolates carried a carbapenemase of KPC, VIM or KPC, reflecting their low prevalence in hospitals.

## Figures and Tables

**Figure 1 antibiotics-11-00917-f001:**
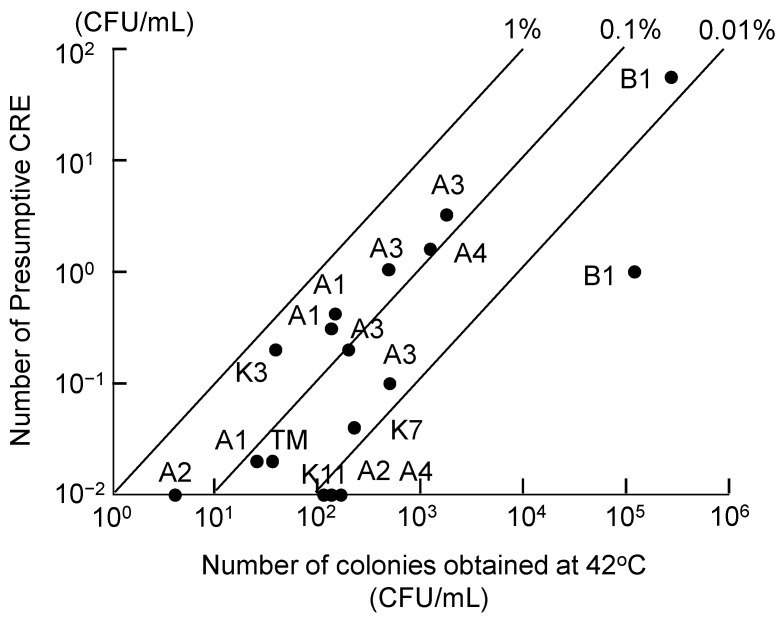
Number of colonies on the plates incubated at 42 °C. Presumptive CRE were also counted from the colonies on plates containing meropenem.

**Figure 2 antibiotics-11-00917-f002:**
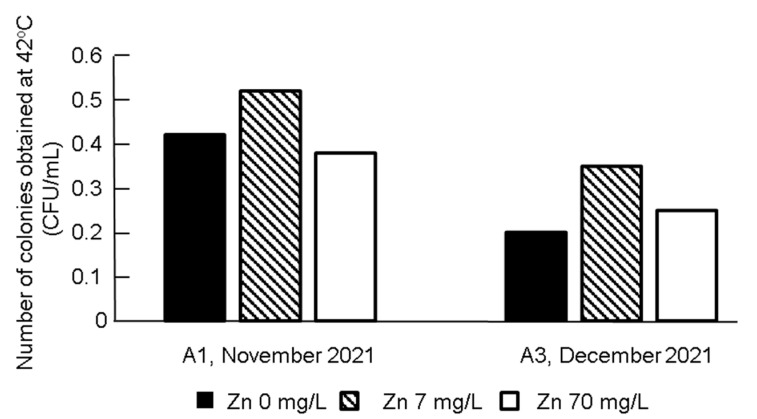
Effect of ZnSO_4_ addition to meropenem-containing culture-medium on the number of colonies.

**Figure 3 antibiotics-11-00917-f003:**
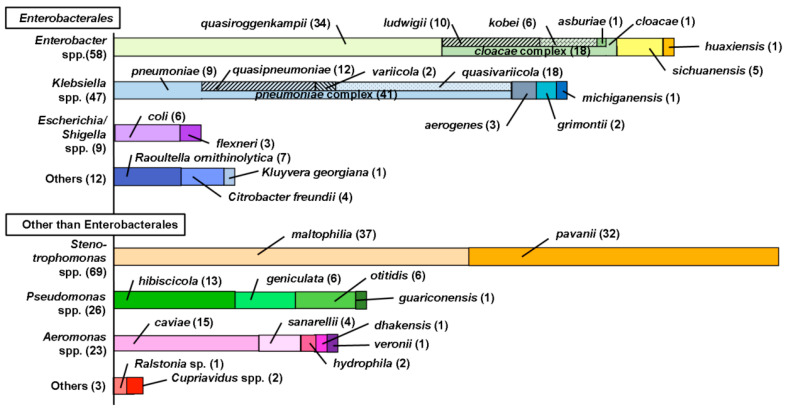
Identified species of presumptive CRE isolates based on 16S rRNA sequencing.

**Figure 4 antibiotics-11-00917-f004:**
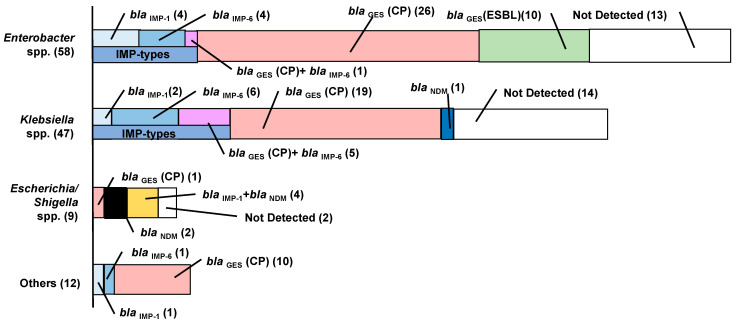
Gene encoding carbapenemase.

**Figure 5 antibiotics-11-00917-f005:**
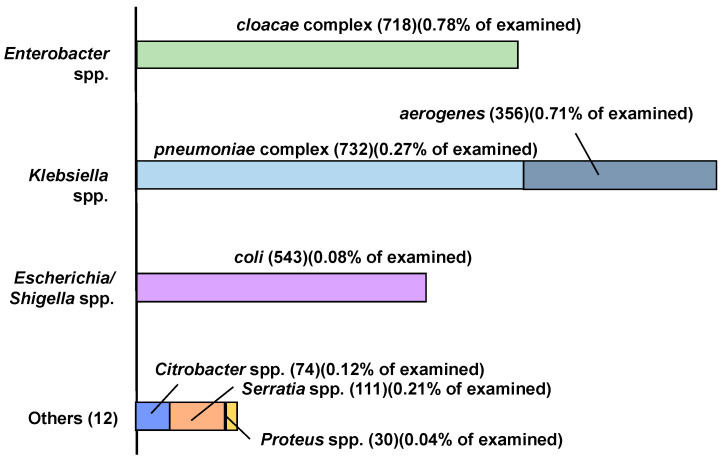
The number of meropenem-resistant Enterobacterales cases reported to JANIS in 2020. Both results on inpatients and outpatients were totaled.

**Figure 6 antibiotics-11-00917-f006:**
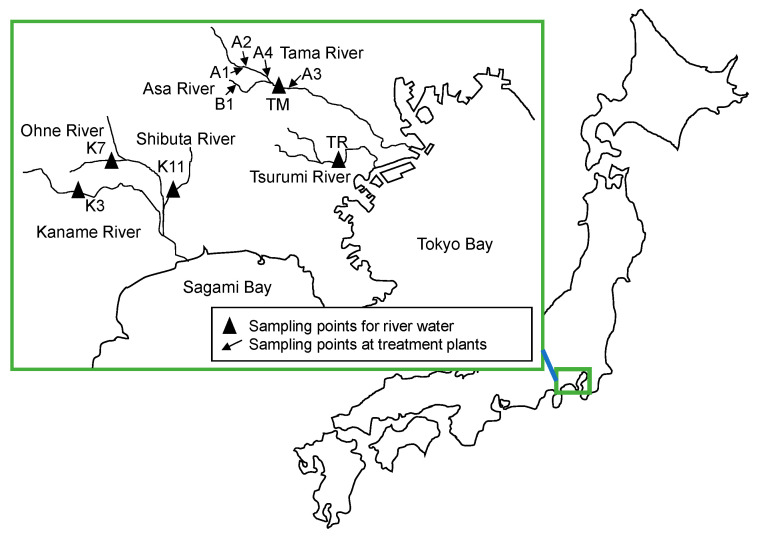
Sampling locations in this study.

**Table 1 antibiotics-11-00917-t001:** Number of isolates examined in this study and the phenotype of the isolates.

		Isolates from 37 °C Plates			Isolates from 42 °C Plates		
		Examined	MPM-R *	MBL-P **	Examined	MPM-R *	MBL-P **
A1	20 August 2019				30	16	11
	25 November 2019	52	24	21	59	39	9
	19 April 2021	30	13	13	7	2	0
	28 November 2021	43	5	1	19	3	0
A2	25 November 2019	2	2	2	3	1	1
	6 July 2020	53	51	48	21	20	16
A3	21 June 2021	35	21	14	20	7	2
	28 July 2021	8	3	1	10	5	4
	18 October 2021	14	12	7	7	5	5
	9 December 2021	21	12	7	5	5	4
A4	12 July 2021	6	6	5	3	2	2
	6 December 2021	0	0	0	1	1	0
B1	26 October 2020	52	48	46	5	4	1
TR	24 September 2019				39	11	13
TM	21 June 2021	5	2	2	1	1	1
K3	24 September 2019				1	1	1
K7	28 October 2021	1	1	1	5	5	3
K11	28 October 2021	45	43	38	2	0	0
Total		367	243	206	238	128	73

*: MPM-R: Meropenem-resistance confirmed, **: MBL-P: production of metallo-β-lactamase confirmed.

**Table 2 antibiotics-11-00917-t002:** Results of identification of genera of meropenem-resistant colonies totaled by each sampling location.

	Enterobacterales					Others					
	Ent.	Kleb.	Esch.	Others	%	Sten.	Pseu.	Aer.	Others	%	Total
River water	0	1	0	0	6	8	6	1	0	94	16
Raw wastewater	0	0	5	0	16	20	4	0	2	16	31
Treated wastewater	58	46	4	12	60	41	16	22	1	40	200
Total	58	47	9	12	51	69	26	23	3	49	247

Ent.: Enterobacter, Kleb.: Klebsiella, Esch.: Escherichia/Shigella, Sten.: Stenotrophomonas, Pseu.: Pseudomonas, Aer.: Aeromonas.

**Table 3 antibiotics-11-00917-t003:** Results of identification of genera of meropenem-resistant colonies totaled by each incubation method.

	Enterobacterales					Others					
	Ent.	Kleb.	Esch.	Others	%	Sten.	Pseu.	Aer.	Others	%	Total
37 °C without Zn	21	13	4	6	32	58	12	22	0	68	136
37 °C with Zn	1	5	0	1	64	3	0	1	0	36	11
42 °C without Zn	33	18	5	3	70	8	14	0	3	30	84
42 °C with Zn	3	11	0	2	100	0	0	0	0	0	16
Total	58	47	9	12	51	69	26	23	3	49	247

Ent.: Enterobacter, Kleb.: Klebsiella, Esch.: Escherichia/Shigella, Sten.: Stenotrophomonas, Pseu.: Pseudomonas, Aer.: Aeromonas.

**Table 4 antibiotics-11-00917-t004:** Results of carbapenemase genes carried by Enterobacterales with different isolation conditions.

	Total	IMP-1	IMP-6	NDM	GES-Type Carbapenemase	GES (ESBL-Like Activity)	% Metallo *
37 °C without Zn	44	4	0	4	27	0	6/44 (14%)
37 °C with Zn	7	0	4	0	4	0	4/7 (57%)
42 °C without Zn	59	7	3	2	24	0	10/59 (17%)
42 °C with Zn	16	0	10	1	7	10	10/16 (63%)
Total	126	11	17	7	62	10	30/126 (24%)

*: percentage of isolates with metallo-β-lactamase genes. The values were not based on the simple total of IMP-1, IMP-6 and MDS because some isolates had both NDM and IMP.

**Table 5 antibiotics-11-00917-t005:** Sample origins for the collection of CRE.

Location	Sampling Dates	Description
A1	20 August 2019, 25 November 2019, 19 April 2021, 28 November 2021	Treated wastewater from municipal large-scale WWTPs (81,000–294,000 m^3^/d)
A2	25 November 2019, 6 July 2020	
A3	21 June 2021, 28 July 2021, 18 October 2021, 9 December 2021	
A4	12 July 2021, 6 December 2021	
B1	26 October 2020	Raw municipal wastewater
TR	24 September 2019	River under the influence of combined sewer overflows and treated wastewater
TM	21 June 2021	
K3	24 September 2019	River under the influence of treated wastewater
K7	28 October 2021	River under the influence of livestock farms
K11	28 October 2021	

## Data Availability

Not applicable.

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
