# Peer review of "Monitoring Carbapenem-Resistant Enterobacterales in the Environment to Assess the Spread in the Community"

_antibiotics, 2022, doi:10.3390/antibiotics11070917_

Round 1
Reviewer 1 Report
The authors present in the manuscript entitled ‘’Monitoring carbapenem-resistant Enterobacterales in the environment for the assessment of the spread in the community’’ a detailed analysis of the species identification and characterization of carbapenem resistance genes obtained from aquatic environments (wastewater, treated wastewater and from river water) in Japan. They also compare their results with the surveillance of carbapenem resistant bacteria from clinical specimens in Japan and they suggest that monitoring of carbapenem-resistant Enterobacterales in the environment may assess the spread in the community. This is an interesting study indicating that reservoirs of CRE may be found in the environment. The methodology is adequate and the presentation of the results is clear and comprehensible. The discussion and the conclusions sections are in accordance with the results section. The references are appropriate.
Major comments:
The Materials and Mathods section may be presented after the Introduction and before the Results section.
Minor comments:
- - please replace ¨total" with ¨a total of" (line 11)
- - please correct ¨specie¨ with "species¨ throughout the document (lines 90, 156, 157, 161, 210)
- - please correct "isolates¨ with "isolate¨ (line 246)
- - please replace "based on a literature" with ¨based on a previous study" (line 336)

Author Response
Reply to Reviewer 1
General comments:
The authors present in the manuscript entitled ‘’Monitoring carbapenem-resistant Enterobacterales in the environment for the assessment of the spread in the community’’ a detailed analysis of the species identification and characterization of carbapenem resistance genes obtained from aquatic environments (wastewater, treated wastewater and from river water) in Japan. They also compare their results with the surveillance of carbapenem resistant bacteria from clinical specimens in Japan and they suggest that monitoring of carbapenem-resistant Enterobacterales in the environment may assess the spread in the community. This is an interesting study indicating that reservoirs of CRE may be found in the environment. The methodology is adequate and the presentation of the results is clear and comprehensible. The discussion and the conclusions sections are in accordance with the results section. The references are appropriate.
Authors' Reply: Thank you for the positive comments for our manuscript.
Major comments:
Comment: The Materials and Methods section may be presented after the Introduction and before the Results section.
Authors' Reply: The authors did not revise the manuscript due to the following reason. The authors followed the word-format provided by the journal official web-site, in which the materials and methods section is positioned after the results and discussion section. For example, a paper published in "Antibiotics" [https://doi.org/10.3390/antibiotics11010048] follows this style.
Minor comments:
- please replace ¨total" with ¨a total of" (line 11)
Authors' Reply: The authors corrected the manuscript according to the suggestion by the reviewer.
- please correct ¨specie¨ with "species¨ throughout the document (lines 90, 156, 157, 161, 210)
Authors' Reply: The authors corrected the manuscript according to the suggestion by the reviewer.
- please correct "isolates¨ with "isolate¨ (line 246)
Authors' Reply: The authors corrected the manuscript according to the suggestion by the reviewer.
- please replace "based on a literature" with ¨based on a previous study" (line 336)
Authors' Reply: The authors corrected the manuscript according to the suggestion by the reviewer.
Reviewer 2 Report
Taro Urase and colleagues (antibiotics-1782633) presented an environmental-based approach to studying carbapenem-resistant bacteria. In general, the manuscript is highly organized and well-written, and the message and implications of this study are a high priority for the field of antimicrobial resistance, environmental hazards and public health. I only have a few minor points.
1. The methodology parameters for enriching carbapenem-resistant bacteria regarding temperature (42) and ZnSO4, should be well discussed.
2. 16S rRNA sequencing is conducted, but the sequencing data is not publicly available.
3. I suggest the quality of figures 3 and 4 should be improved, please make them colour figures.
4. Sample origins, and geographic map should be given for detailed samplings.
5. MIC value for individuals is not available, and genomic sequencing for individual bacterial isolates could deliver much more comprehensive knowledge for this study.
6. One limitation is that only those established IMP, NDM, VIM, KPC, OXA, GES gene was studied. WGS is needed for expanding our knowledge in this field.
Author Response
Reply to Reviewer 2
Taro Urase and colleagues (antibiotics-1782633) presented an environmental-based approach to studying carbapenem-resistant bacteria. In general, the manuscript is highly organized and well-written, and the message and implications of this study are a high priority for the field of antimicrobial resistance, environmental hazards and public health. I only have a few minor points.
Authors' Reply: Thank you for the positive comments for our manuscript.
- The methodology parameters for enriching carbapenem-resistant bacteria regarding temperature (42) and ZnSO4, should be well discussed.
Authors' Reply: The authors newly prepared table 4 for the carbapenemases genes carried by Enterobacterales with different isolation conditions. To explain this table, one paragraph was added to section 2.4. In addition, a section 3.3 (Effect of isolation condition on the detected type of carbapenemase) was added in the revised manuscript.
- 16S rRNA sequencing is conducted, but the sequencing data is not publicly available.
Authors' Reply: The objective of sequencing 16S rRNA in this study is to find the closest accession to identify the species of the isolates. We considered that the registration of the sequencing results of 16S rRNA is not needed, because we do not discuss in detail for the genetic differences of the isolates. To partly comply with the authors suggestion, we added the closest accession numbers for isolates which have to be treated carefully for the identification of the species, wring in section 2.3 " The five isolates identified as E. sichuanensis (the closest accession POVL01000141) had only 6 - 9 variations in more than 1460 nucleotide sequences compared with those of E. ludwigii (accession JTLO01000001) or E. Kobei (accession CP017181). Only one isolate identified as E. huaxiensis (the closest accession MK049964) had less similarity than 98.65% to any accessions of known E. cloacae complex isolates. These results indicated that most of the isolates identified as Enterobacter spp. in this study belonged to E. cloacae complex and the related species." and " Seven isolates were identified as Raoultella ornithinolytica (the closest accession AJ251467). These seven isolates had relatively small numbers (13-17) of variations in more than 1460 nucleotide sequences (more than 98.9% -99.2 % similarity) compared with those of K. aerogenes (Accession CP002824). " Bacause of the small difference in 16S rRNA between K. aerogenes and Raoultella ornithinolytica, we added in the discussion 3.1 " It must be noted that the isolates identified as Raoultella ornithinolytica in this study might be identified as K. aerogenes in the routine clinical examination considering the small difference (98.9% -99.2 % similarity) in 16S rRNA sequencing results." In the conclusion, a small change in the same context was added.
- I suggest the quality of figures 3 and 4 should be improved, please make them colour figures.
Authors' Reply: The authors prepared colored Figures 3, 4 and 5.
- Sample origins, and geographic map should be given for detailed samplings.
Authors' Reply: The authors newly prepared Figure 6.
- MIC value for individuals is not available, and genomic sequencing for individual bacterial isolates could deliver much more comprehensive knowledge for this study.
Authors' Reply: The authors agree with the reviewer comments that the MIC value for individual isolates may be related to the characteristics (phenotype and genotype) of the isolates. However, we measured only inhibition zone diameters in this study and we did not analyze the relation between the inhibition zone diameter and the characteristics of the isolates. We did not revise the manuscript on this point, because it may be difficult for the authors to improve the content.
- One limitation is that only those established IMP, NDM, VIM, KPC, OXA, GES gene was studied. WGS is needed for expanding our knowledge in this field.
Authors' Reply: The authors agree with the reviewer comments that more intensive analysis of carbapenemases genes is needed. The authors consider that the objective of this study is limited to the clarification of species and genotypes of carbapenem-resistant Enterobacterales monitorable from the environments for the possible application of wastewater-based epidemiology. We did not revise the manuscript on this point, because we consider that WGS will be the next step of our study.